# A light tunable differentiation system for the creation and control of consortia in yeast

Chetan Aditya [1,2,3], François Bertaux [1,2], Gregory Batt [1,2,4 ✉] & Jakob Ruess[1,2,4 ✉]

Artificial microbial consortia seek to leverage division-of-labour to optimize function and possess immense potential for bioproduction. Co-culturing approaches, the preferred mode of generating a consortium, remain limited in their ability to give rise to stable consortia having finely tuned compositions. Here, we present an artificial differentiation system in budding yeast capable of generating stable microbial consortia with custom functionalities from a single strain at user-defined composition in space and in time based on optogenetically-driven genetic rewiring. Owing to fast, reproducible, and light-tunable dynamics, our system enables dynamic control of consortia composition in continuous cultures for extended periods. We further demonstrate that our system can be extended in a straightforward manner to give rise to consortia with multiple subpopulations. Our artificial differentiation strategy establishes a novel paradigm for the creation of complex microbial consortia that are simple to implement, precisely controllable, and versatile to use.

[1] Inria Paris, 2 rue Simone Iff, 75012 Paris, France. [2] Institut Pasteur, 28 rue du Docteur Roux, 75015 Paris, France. [3] Université de Paris, 85 Boulevard Saint-Germain, 75006 Paris, France. [4] These authors contributed equally: Gregory Batt, Jakob Ruess. ✉email: gregory.batt@inria.fr; jakob.ruess@inria.fr

The evolutionary transition from single cell to multicellular organisms marked a critical turning point in biology[1]. Such shift relied on optimizing fitness and productivity through division of labour and specialization[2,3]. The same principle can be extended to microorganisms living together to form microbial communities or consortia. Engineered microbial consortia hold enormous potential and have been hailed as the next frontier in synthetic biology[4,5]. Proof of concept studies have concretely established applications in bioproduction[6,7], bioremediation[8,9], and soil microbiome engineering[10], paving the way for therapeutic applications using human microbiome engineering[11,12].

In the context of bioproduction, microbial consortia possess several advantages over traditional monocultures as functional specialization allows metabolic burden to be shared across different species. Diversification thus allows yields to be optimized simply through tuning consortia composition, rather than re-engineering the strain itself[13]. Moreover, by including multiple species, toxic by-products produced by one species can be sequestered and/or metabolized by another, thereby improving the efficiency of the overall process[14]. Microbial consortia are typically generated by culturing two or more species together. Such co-culturing approaches rely on various inter-species interactions to ensure the co-existence of different species like mutualism[15], emergent cooperation[16], competitive amensalism[17], and predation[18]. Despite considerable advances in our ability to engineer microbial consortia[6,8,9,19–21] and in our understanding of community interactions[15,16,18,21,22], dynamic control of consortium composition remains a key challenge in the field[19]. Typically, stable consortia are based on syntrophic or quorum sensing interactions that, albeit being autonomous, remain critically dependent on cell density, thus limiting the applicability for dynamic control. Additionally, scaling the consortium to include more than two species requires non-trivial considerations that may not lead to stable co-existence[20]. In light of these limitations, an externally controllable differentiation system could be well suited to address this challenge.

In recent years, advances in biological control have come from coupling computers with growing cells carrying the engineered system, made possible by special platforms that integrate biological systems with the computer via a feedback loop[23–27]. The development of optogenetics, i.e. the use of light to trigger cellular processes, has contributed significantly to control applications by increasing the spatiotemporal resolution of the control signal[23,24,28–38]. Control of protein expression using light has been demonstrated both at the population level[23,28,31] and in single cells[30,33,34,37]. Optogenetics has been used to control cellular processes in other contexts, for instance, signalling dynamics[24], morphogenesis[36], neuroscience[38], bioproduction, and metabolic engineering[29,35]. However, control of population dynamics using optogenetics in a multispecies environment has not been demonstrated yet.

Here, we present an artificial differentiation system in *S. cerevisiae* capable of generating a microbial consortium composed of functionally different subpopulations emerging from a single population akin to differentiation in multicellular organisms. Concretely, we achieve differentiation into genetically distinct subpopulations—henceforth referred to as species to highlight the analogy to natural microbial consortia—via recombination-based genetic rewiring that can be externally controlled via light. We demonstrate that our system shows desirable features including low background activity, high efficiency for optogenetic recombinases in budding yeast, graded response to varying light signals, absence of hysteresis, and dynamics that are fast, predictable, and tunable. The system reaches >99% differentiation after 4 h of light stimulation and can be stably maintained at any given intermediate level of differentiation for long periods of time (>48 h).

Owing to its fast and predictable dynamics, our differentiation system enables rapid and robust bidirectional control of a microbial consortium arising from a single strain at user-defined compositions in continuous cultures for extended periods in dynamic setups. Coupling the system to a growth arrest module allows us to control population growth rates in continuous culture in different physiological contexts. We show that our system can be extended to give rise to complex multispecies microbial consortia. We engineer two differentiation programmes that can be used to control the total number of species. Finally, we show that our system allows for spatial structuring of microbial consortia by imprinting patterns in 2D cultures with high resolution. To the best of our knowledge, this is the first report of light-driven system for control of a microbial consortium.

## Results

**An optogenetic synthetic differentiation system in *S. cerevisiae*.** We constructed an optogenetic differentiation system consisting of a blue light-inducible Cre recombinase under the control of a constitutively expressed optogenetic transcription factor, EL222 (refs. [39,40]). In order to test the functionality of the system, we designed a recombination cassette composed of a floxed coding sequence (CDS), coding for a fluorescent reporter (mCerulean) that is transcribed constitutively via a pTDH3 promoter upstream of the first LoxP site. Another CDS, coding for a different fluorescent reporter (mNeonGreen), was added downstream of the second LoxP site (Fig. 1a, top and Supplementary Fig. 7).

Prior to differentiation, mCerulean is constitutively expressed and mNeonGreen is not. After light induction, Cre is expressed causing a recombination event leading to the expression of mNeonGreen and loss of mCerulean expression.

**Fast and efficient differentiation in light, low background activity in the dark.** In order to evaluate efficiency and background activity, cells harbouring the differentiation system were cultured to the exponential phase in batch and then subjected to either blue light or kept in darkness. Samples were taken at regular intervals and passed through the cytometer (Fig. 1a, bottom). We used flow cytometry data to compute the differentiated fraction by applying a threshold on mNeonGreen fluorescence (Fig. 1b). We observed only a marginal increase in the differentiated fraction after 72 h of culture in the dark suggesting low background activity (Fig. 1c, grey inset). Induction with blue light triggered differentiation. Moreover, the efficiency of the system in budding yeast was superior when compared to existing light inducible systems in *S. cerevisiae*, leading to >99% differentiation after 4 h of induction (Fig. 1c, blue inset) (Supplementary Table 2)[41–44]. The efficiency of our system allows us to achieve high levels of differentiation with minimal light exposure thus eliminating the risk of phototoxicity (Supplementary Note 3 and Supplementary Fig. 8).

Dynamics of the recombined fraction could be modulated by varying either the light intensity or the duration of applied light pulses. We note that the differentiated fractions are reminiscent of EL222 inducible fluorescent protein levels for varying light intensities or the frequency or duration of applied light pulses (Supplementary Note 3 and Supplementary Figs. 9 and 10)[40].

To further establish that our system remains functional in different experimental contexts, we cultured cells in a microfluidic chamber and stimulated them periodically via short pulses of light on our microscopy platform[30]. Through regular imaging, cellular fluorescence was used to classify cells as differentiated (Fig. 1d). Prior to light induction (Fig. 1d, $t = 0$) less than 2% of the cells were differentiated ($n = 817$, over eight fields from two independent experiments) and within 8 h of induction more than

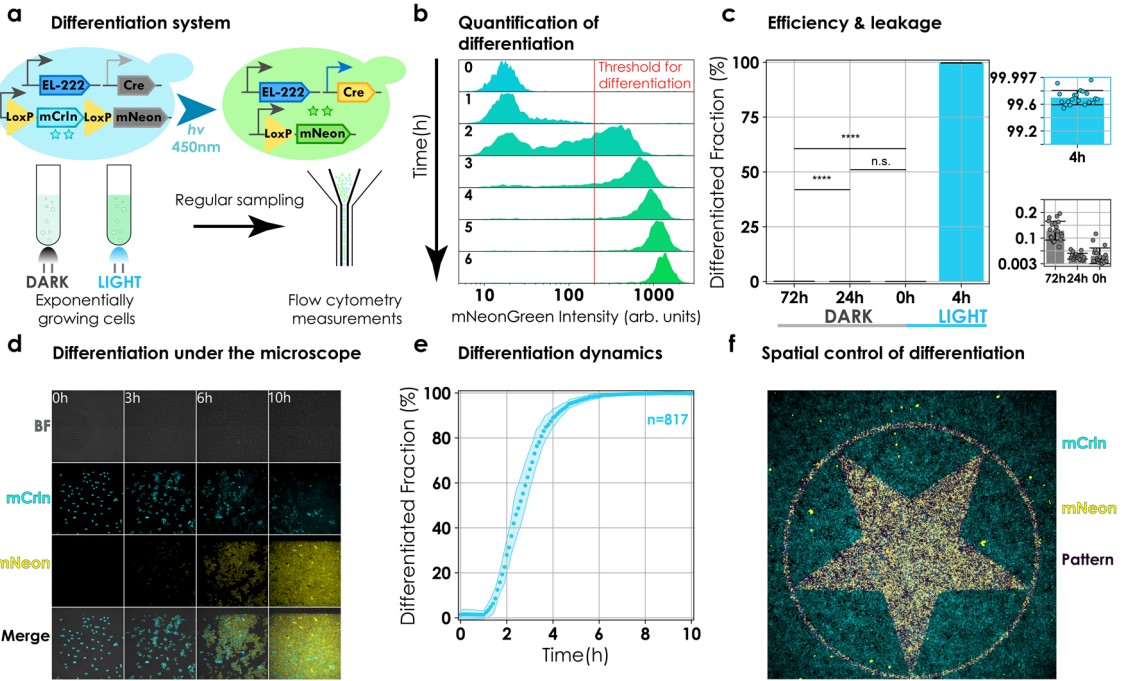

**Fig. 1 Design and functional characterization of the differentiation system in liquid and 2D cultures. a** Design and experimental setup. Cells constitutively express EL222 optogenetic transcription factor. Blue light triggers expression of Cre and recombination, and a change in fluorescence from mCerulean to mNeonGreen. Cells carrying the differentiation system were cultured to exponential phase in batch and induced via LEDs. Cytometry measurements were made by sampling at regular intervals. **b** Quantification of differentiation. Cells are classified as differentiated if cellular mNeonGreen fluorescence exceeds 200 arb. units (red line). The plot shows the evolution of population mNeonGreen fluorescence as a function of time. The threshold was set such that only cells that have expressed detectable amounts of mNeonGreen are classified as differentiated. **c** Efficiency and background differentiation. Twenty colonies were picked and cultured in batch for 72 h in dark or induced at $t = 0$ for 4 h. Measurements were taken at $t = 0$, $t = 24$ h and $t = 72$ h. Bars represent means from a single experiment. Error bars signify standard deviation. Individual data points from colonies are depicted in a scatter plot overlaid on the bar plot (blue and grey insets). Increase in differentiated fraction was not significant (n.s.) after 24 h ($p = 0.41$) but became significant at 72 h ($p = 5.3e-08$; two-sided paired $t$-test) (black horizontal lines). **d** Snapshots of cell growth and differentiation under the microscope. Images during induction from a representative field of view. Cellular fluorescence changes from mCerulean (cyan) to mNeonGreen (yellow) (Supplementary Movie 1). **e** Differentiation dynamics under the microscope. Images were segmented and analysed. To be deemed differentiated, median cellular fluorescence had to exceed 300 arb. units mNeonGreen fluorescence. Circles represent mean differentiated fraction over eight fields of view from two independent experiments. Shaded region shows standard error of mean. The total number of cells at $t = 0$ summed over all fields of view are given by $n$ ($n = 817$). **f** Imprinting patterns in the population. Cells were allowed to form a monolayer inside a μIbidi slide. A user-defined pattern was illuminated over the monolayer using a digital mirror device (DMD). Merge consists of mCerulean fluorescence (cyan), mNeonGreen fluorescence (yellow), and the pattern (magenta).

99% of the cells in the field of view had differentiated (Fig. 1d, $t = 6$ h and Supplementary Movie 1). The differentiation dynamics were reproducible (Fig. 1e and Supplementary Fig. 6).

**Spatial control of differentiation and pattern formation in 2D cultures**. To control population composition in space, we grew cells harbouring the differentiation system to the late exponential phase. Cells were then loaded in a μIbidi slide and placed under the microscope. We used our microscopy platform equipped with a digital micromirror device to periodically shine blue light in the shape of different patterns (Supplementary Fig. 20). Cells were illuminated with a given pattern for 1 s every 3 min during 1 h (Supplementary Note 6). Following this, cells were kept in darkness for an hour prior to imaging to ensure a good assessment of the differentiation state of cells (time for the mNeonGreen protein to be produced and mature). We observed that accurate patterns of differentiated cells emerged (Fig. 1f). Some recombination was present outside of the provided pattern, but it is very likely that these cells had differentiated long before the start of the experiment since they lacked mCerulean fluorescence. We conclude that our optogenetic differentiation system enables precise spatial control of differentiation, suggesting that it is a practical tool to generate spatially structured heterogeneous microbial communities composed of functionally distinct subpopulations.

**Characterization of the system and development of a predictive model**. To characterize the differentiation behaviour, cells were cultured continuously in the exponential phase in our LED-equipped custom turbidostat platform[28] and induced with different light inputs (Fig. 2a). Sampling from the culture and flow cytometry measurements were automated ("Methods"). We observed that, by modulating duration of light pulses (Fig. 2a, top), the system could be regulated to reach intermediate levels of differentiation that were stable over time (>48 h, Fig. 2b), showing that we can stably maintain microbial consortium at different compositions (Fig. 2c). Moreover, these results reveal an interesting dichotomy of our system: it is capable of eliciting a graded response to different stimuli at the population level as well as a differential response to the same stimulus at the single-cell level (Figs. 1d and 2c).

Next, we wondered if hysteresis is present in our system given repeated stimuli. To gauge the extent to which hysteresis affects differentiation dynamics, we induced cultures with repeated

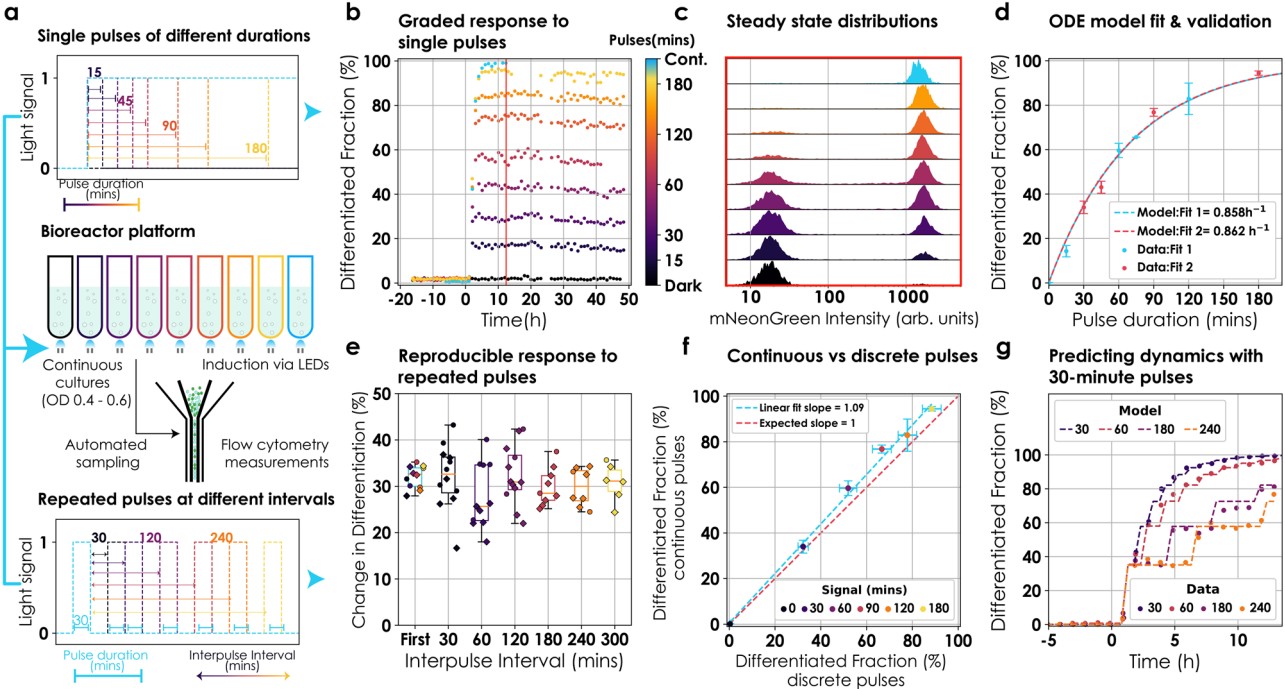

**Fig. 2 Characterization and modelling in continuous cultures. a** Bioreactor platform and induction profiles. Cells were cultured continuously in the exponential phase using our bioreactor platform[28]. Cultures were induced via LEDs and flow cytometry measurements were automated. Induction was in the form of either single pulses of different durations (top) or repeated pulses of 30 min at different interpulse intervals (bottom; only the first two pulses are represented). **b** Differentiation dynamics after single pulse induction. Cultures were induced at $t = 0$ with single pulses of light ranging between 15 and 180 min (colour bar). Following induction, cultures were kept in the dark for 48 h. Circles represent values from a unique experiment. **c** Tunable population composition. Snapshots of population mNeonGreen fluorescence after induction with single pulses (**b** red line indicates the time of each snapshot). **d** Model fitting and validation. An ODE model was fitted to single pulse induction data. Circles represent mean steady-state differentiation fractions of three independent experiments (except 75-min pulse, unique experiment). Error bars signify s.d. Blue and red circles were independently used to fit the ODE model (dashed lines). **e** Reproducible behaviour with repeated pulses. Cultures were stimulated with 30-min pulses repeated at different interpulse intervals (30–300 min). Circles and diamonds represent the change in differentiation fraction by individual pulses from two independent experiments. Data were collated over the two experiments for boxplots. The colour of circles and boxplots reflects interpulse interval. The first pulse of each experiment was used for the blue boxplot. Lines, boxes, and whiskers denote median, quartiles, and extreme values, respectively. **f** No observable memory effect. Data from (**d** and **e**) were used to the compare the efficiency of continuous light vs discrete pulses. Circles represent differentiation effected by continuous pulses ($y$ axis) and equivalent duration of induction in form of 30-min pulses ($x$-axis). Data are presented as mean values ± s.d.; $n = 12$ for discrete pulses and $n = 3$ for continuous pulses, where $n$ is the number of pulses delivered. A linear fit of the data is given by the blue dashed line and compared to expected linear behaviour in the absence of memory (red line). **g** Predicting dynamics. Data (circles) from (**e**) was used to check the predictability of the ODE model (dashed lines). Induction started at $t = 0$. Model predictions were shifted in time to account for observation delay.

pulses at various interpulse durations (Fig. 2a, bottom). If hysteresis were present, we would expect smaller population fractions to recombine with subsequent pulses compared to the first one. The response was reproducible with each pulse resulting in the same differentiated fraction regardless of prior exposure to light (Fig. 2e) (up to small reactor-to-reactor variability). Furthermore, we investigated if time-separated, repeated pulses of a given total duration would result in a higher differentiation fraction compared to a single continuous pulse of same duration. We found that continuous light resulted in similar differentiation fractions as discrete pulses for the same total duration of induction (Fig. 2f).

To be able to predict the differentiated population fraction for a given light input, we developed an ordinary differential equations (ODE) model with a single parameter, the differentiation rate (Supplementary Note 3). This model was fitted to two datasets of steady-state differentiation fractions post induction with mutually exclusive single pulses. Both fits captured observed behaviours well and resulted in similar estimates for the differentiation rate (Fig. 2d). A full description of the model as well as our fitting strategy are presented in the Supplementary Note 3. We also used the model to predict differentiation

dynamics emerging from additional light input sequences and found that model predictions were in good agreement with observed data (Fig. 2g). We conclude that the differentiation dynamics of our system is predictable and can be captured by a simple ODE model.

**Robust dynamic control of heterogeneous synthetic communities emerging from a single strain.** Having established that our differentiation system exhibits fast and predictable dynamics, we investigated if we could employ it to control consortia composition at user-defined levels in a dynamic setting. We contrived an experimental setup in which two reactors were coupled such that the output of one reactor was connected to the input of the other to be able to dynamically control the population composition despite the absence of growth rate differences between differentiated and non-differentiated cells. The first reactor was kept in the dark as a 'reservoir' for non-differentiated cells and the second 'control' reactor was exposed to light signals to maintain the culture at a target set point for the differentiated fraction. Cytometry samples were taken every hour to observe the state of the control reactor and adjust the control signal (Fig. 3a). Cultures in

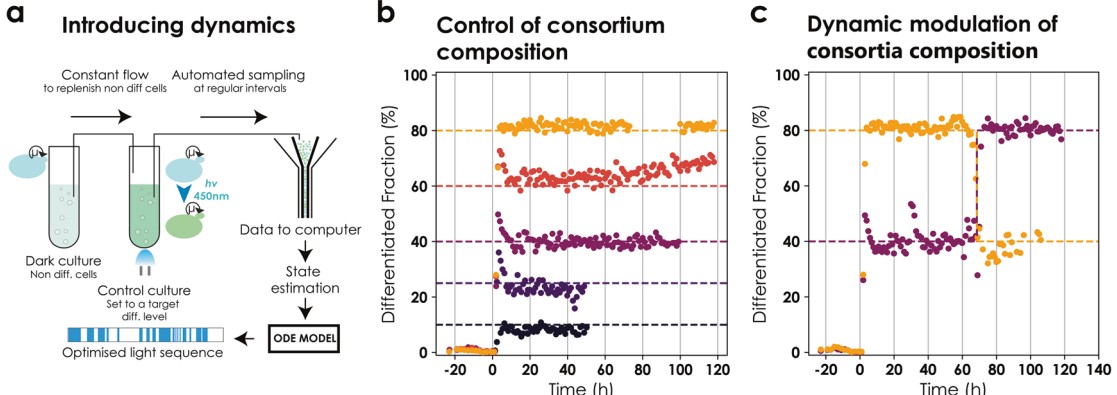

**Fig. 3 Dynamic control of population composition in a microbial consortium. a** Setup for two reactor control. Cells harbouring the original differentiation system were continuously cultured at fixed but different cell densities in two reactors simultaneously. The first reactor was kept in the dark as a reservoir of non-differentiated cells. The output of this vessel was connected to the second 'control' reactor. The control culture was set to a target level of differentiation and continuously monitored via automated flow cytometry measurements that were analysed online. The system state was estimated from analysed data and sent to the model predictive control (MPC) module. The MPC module provided an optimized light sequence to maintain the culture at the desired set point (Supplementary Note 5). **b** Control of consortium composition. Cultures were targeted to 10–80% differentiation. Control started at $t = 0$. Circles signify differentiated fractions. Each colour corresponds to a different control experiment and the dashed line reflects the target set point. Note that the figure is composed of independent experiments of different lengths. Data were removed when the OD, either in the reservoir or the control reactor, could not be maintained at the desired target. Light signals are provided in Supplementary Fig. 17. **c** Bidirectional control of consortium composition. Cultures were targeted to 40 and 80% differentiation. Data are represented as in (**b**). The target was changed at $t = 60$ h to 80% and 40%, respectively.

both the reservoir and control reactors were maintained at constant cell densities (Supplementary Note 5). In addition to the feed from the reservoir reactor, the control reactor was fed fresh media to maintain the culture at the target cell density. Since there was a constant flux of non-differentiated cells from the reservoir to the control reactor, we adjusted the ODE model accordingly.

Using a model predictive control (MPC) framework[28] along with the modified ODE model (Supplementary Note 5 and Supplementary Fig. 16), we attempted to maintain user-defined consortia composition (differentiation fractions) in the control reactor. The framework consisted of sampling cells from the control reactor at regular intervals, performing online data analysis to estimate the state of the system (fraction of differentiated cells), and updating the light signal to maintain the fraction of differentiated cells at the desired set point (Supplementary Note 5). We were able to control the population compositions in the control reactor and maintain them for extended periods (up to 96 h) (Fig. 3b) for different target set points ranging from 10 to 80% of differentiated cells. The response was quick and the desired population fractions were reached within approximately 6 h of starting the control. Additionally, our setup enabled us to dynamically modulate the population composition in both directions, i.e., increase or decrease differentiated cells in the population from a given level. Concretely, we targeted two cultures to 40% and 80% differentiated cells in the population, respectively and changed the set points after 60 h to 80% and 40%, respectively (Fig. 3c). The control reactor required active control and lost the desired population composition in the absence of an appropriate light signal.

To eliminate the need for a reservoir of non-differentiated cells, we sought a strategy that would allow the non-differentiated fraction to be replenished over time (non-differentiated cells enrich in absence of light signal). To this end, we coupled our system to a growth arrest module such that cells growth arrest upon differentiation (GAuDi), forcing them to be outcompeted by the non-differentiated cohort after a single exposure to light.

To achieve the coupling, we hijacked the mating factor pathway in *S. cerevisiae* by overexpressing a FAR1 variant after

differentiation[45]. FAR1, a CDK inhibitor, is the downstream effector of the mating factor growth arrest and arrests the cells in G1–S transition by blocking the interaction between CDC28 and G1 cyclins[46,47]. Briefly, differentiated cells express ATAF1, an orthogonal transcription factor[48], that, in turn, drives the overexpression of the FAR1 variant. A positive feedback loop on ATAF1 TF leading to higher expression of the FAR1 variant was necessary to obtain effective growth arrest upon differentiation. This came at the cost of higher leakage in the dark. The complete construction of this strain is depicted in Supplementary Fig. 12.

We characterized the GAuDi strain under the microscope and in continuous cultures (Supplementary Note 4, Supplementary Fig. 13 and Supplementary Movie 2) and concluded that our approach allowed us to optogenetically induce growth arrest (Fig. 4b, c) and replenish the non-differentiated fraction in self-contained configuration after transient light induction (Fig. 4b and Supplementary Fig. 14). We observed that under the microscope, growth arrest was complete, and no cell divisions were detected upon differentiation for 15 h (Supplementary Movie 2). Similarly, in the presence of continuous light, the growth rate in liquid cultures decreased dramatically (<0.04 h⁻¹) (Supplementary Fig. 13). In the turbidostat, we observed that the cultures under continuous light escaped the growth arrest 15–20 h post differentiation. We establish by targeted genome sequencing and culture in selective media that the escape is linked to loss of the integrative cassette (Supplementary Fig. 13). We also observed a subpopulation that possessed neither green nor red fluorescence and were deemed dead. We adapted the ODE model to account for growth arrest and extended it to include cell death and escape. We assumed that differentiated cells, in addition to growing significantly slower, die and escape from the growth arrest at definite rates. The culture growth rate is then given by a weighted average of the growth rates of individual species and is equal to the dilution rate of the reactor at constant cell density. The model was fitted to dynamical data from an experiment with a non-trivial light signal comprised of pulses of varying durations

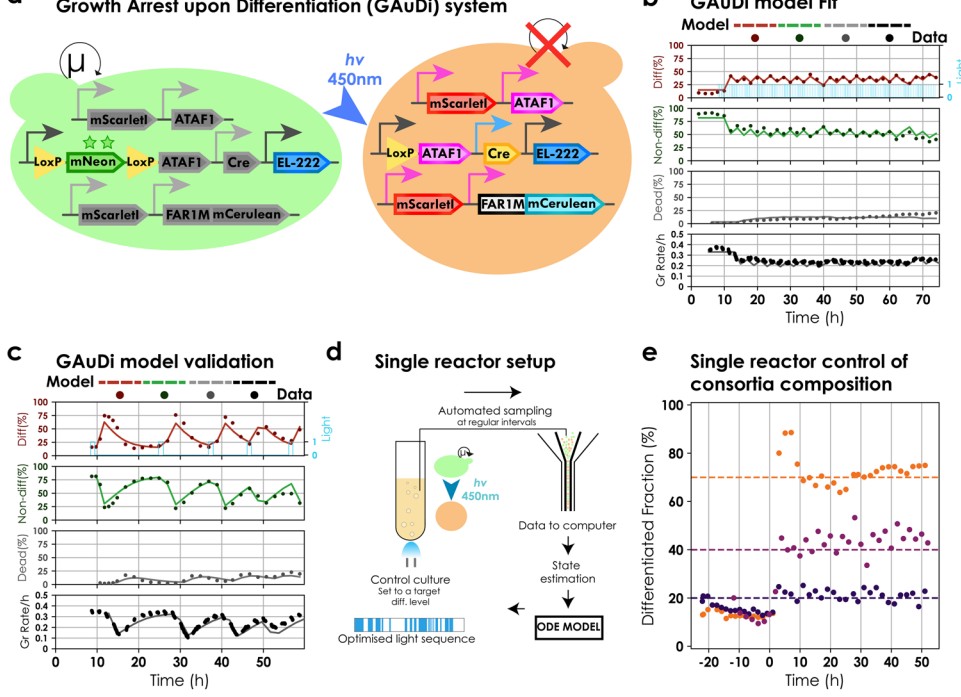

**Fig. 4 Characterization and modelling of GAuDi. a** Design. GAuDi cells constitutively express EL222. Blue light triggers expression of Cre and recombination, and expression of ATAF1 transcription factor. ATAF1 drives the overexpression of a FAR1 mutant (that arrests cells in G1) and mScarlet-I, a "marker" for differentiation. An additional copy of ATAF1 driving itself was necessary for effective arrest. **b** GAuDi ODE model fit. An ODE model for GAuDi was developed phenomenologically. The model was fitted to dynamical data with repeated light stimuli of different durations. Solid circles represent data and solid lines signify model predictions. We note that before $t = 0$, a persistent non-zero differentiation fraction was present possibly due to leakiness in ATAF1 expression in a subset of the population (Supplementary Note 4). **c** GAuDi model validation. The fitted model was validated by predicting dynamic responses to light signals that were not used for the fit. Data are represented as in (**b**). Globally, model predictions were in good agreement with the data (Supplementary Fig. 15). **d** Setup for single reactor control. Cells harbouring the GAuDi system were continuously cultured in the exponential phase. The culture was set to a target level of differentiation and continuously monitored via automated flow cytometry measurements that were analysed online. The system state was estimated from analysed data and sent to the model predictive control (MPC) module. The MPC module provided an optimized light sequence to maintain the culture at the desired set point (Supplementary Note 5). Due to presence of dead cells, data were filtered and only live cells were used for subsequent analysis. **e** Single reactor, single strain control of a microbial consortium. Cultures were targeted to 25, 40, and 70% differentiation. Control started at $t = 0$ h. Circles signify differentiated fractions. Each colour corresponds to a different control experiment and the dashed line reflects the target set point.

at different intervals (Fig. 4b and Supplementary Note 4). The fitted model was validated by comparing model predictions to data collected in four additional experiments in which cultures were exposed to different light signals (Fig. 4c and Supplementary Fig. 15b). We note that the predictive power of the model is limited when the system operates under strong selection pressure, that is, when light is applied over extended durations.

We used this modified ODE model in conjunction with the MPC framework to demonstrate single reactor control of a microbial consortium originating from a single strain. Concretely, exponentially growing GAuDi cells in the turbidostat were exposed to optimized light signals such that the population composition is maintained at user-defined set points. Flow cytometry measurements were taken at regular intervals to track the fraction of differentiated cells (Fig. 4d). We found that population compositions could be maintained stably for extended periods (up to 48 h) (Fig. 4e). We note that due to genetic stability limitations, it was possible to maintain composition control for longer periods only if the target set point was below 50% differentiation. Evolutionary constraints are known to limit the long-term stability of synthetic circuits[49] particularly circuits that are engineered to implement growth arrest/self-killing at the population level[18,50]. To the best of our knowledge, this is the first report of dynamic control of population composition in a

two species artificial microbial consortium arising from a single strain.

**Generation of complex microbial consortia with multiple subpopulations**. Certain applications, particularly in metabolic engineering, might require microbial consortia composed of more than two species. To probe if our system can be used to engineer differentiation programmes that allow one to create complex multispecies consortia, we cloned two recombination cassettes (denoted by C and N) in a single strain along with the differentiation system (Fig. 5a). Cells were cultured and induced in our bioreactor platform with varied pulses repeated every 360 min. We found that both sites were approximately equally likely to recombine (asynchronous) (Fig. 5b) resulting in a four species microbial consortium, consisting of double recombined cells ($\overline{CN}$), single recombined cells ($C\overline{N}$, $\overline{C}N$), and the original non-recombined population (CN).

Noting that the length of the to-be-excised region plays a critical role in determining the differentiation rate (Supplementary Fig. 21), we set out to test if this could be exploited to modify the differentiation dynamics. We cloned two recombination cassettes of unequal to-be-excised region (C and N′) along with the differentiation system (Fig. 5c). We observed that the shorter site recombined first ($\overline{C}N'$) and subsequent pulses led to

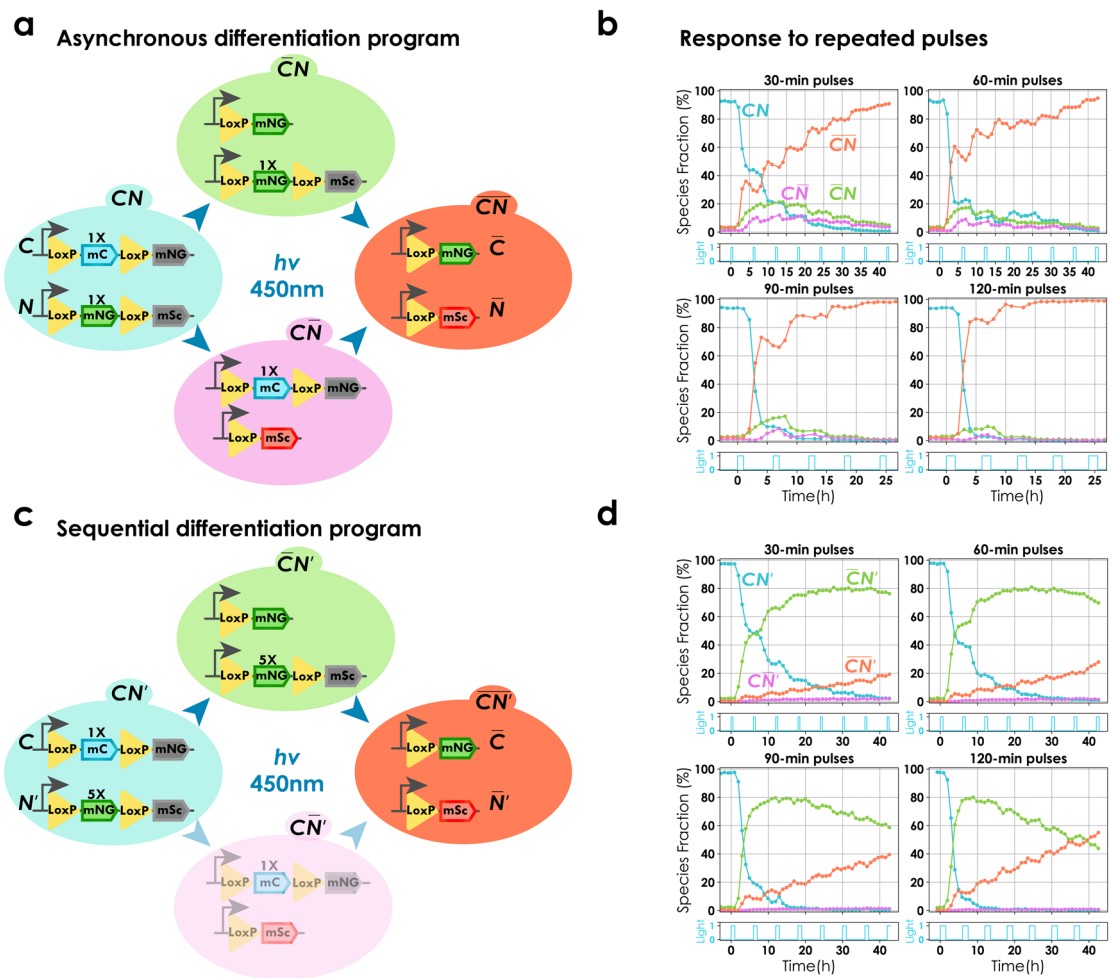

**Fig. 5 Engineering differentiation programmes for complex consortia with multiple subpopulations. a** Asynchronous differentiation programme. Two recombination cassettes (C and N) were introduced along with the differentiation system in the same strain. Both cassettes consisted of to-be-excised region of equal lengths (1× of the original system (Fig. 1a). All four possible cell types are shown (neither cassettes recombined (CN), single cassette recombined (C̄N & CN̄), and both cassettes recombined (C̄N̄)). Circuits shown in the panel are representational (Supplementary Fig. 22). **b** Response to repeated pulses of different lengths. Four cultures were induced with repeated light pulses of different durations, 6 h apart. Each subplot represents data from a single experiment with unique pulse duration given by cyan subplots (clockwise from top right, 30, 60, 90, and 120 min, respectively). Solid lines show dynamics of prevalence of each subpopulation (blue, purple, red, and orange for CN, C̄N̄, CN̄, and C̄N, respectively). Circles indicate individual time points. Prevalence of each population was calculated using thresholds on mNeonGreen, mScarlet-I, and mCerulean fluorescence (Supplementary Note 7 and Supplementary Fig. 23). All four subpopulations could be observed. We note that the small decrease in the fraction of C̄N cells just after differentiation and a corresponding increase in CN is an artefact of the threshold-based quantification that is limited by the dilution of mNeonGreen. **c** Sequential differentiation programme. Two recombination cassettes were introduced along with the differentiation system in the same strain. Relative length of the to-be-excised region of one cassette (N′) was 5× longer than the other (C). All four possible cell types are shown. Notations resemble **a**. Due to increase in the length of to-be-excised region, the differentiation rate of N′ was significantly slower than C leading to inappreciable prevalence of CN̄′ (faded cell type) (Supplementary Fig. 22). **d** Response to repeated pulses of different lengths. Four cultures were induced with repeated light pulses of different durations, 6 h apart. Each subplot represents data from a single experiment with unique pulse duration (clockwise from top right, 30, 60, 90, and 120 min, respectively). Data are represented as in (**b**). Three subpopulations could be observed.

enrichment of cells with both sites recombined (C̄N̄′) (Fig. 5d) constituting a sequential differentiation programme as opposed to the asynchronous programme obtained with to-be-excised regions of equal lengths. We did not observe appreciable levels of species with only the longer site recombined (CN̄′) effectively resulting in a three species microbial consortium.

Both versions of the system with two recombination sites remained capable of eliciting a graded response to light. These results demonstrate that our differentiation system can be extended to implement distinct differentiation programmes capable of yielding complex consortia composed of a controllable number of species whose prevalence can be optogenetically tuned.

## Discussion

Microbial consortia are expected to be of great utility for biotechnology and hold immense potential for diverse applications[4–11]. However, dynamic control of consortium composition remains relatively unexplored despite being a key challenge in the field[19]. In the present study, we address this challenge with the help of an artificial differentiation system in *S. cerevisiae* capable of generating microbial consortia with custom composition. The system is based on blue light inducible expression of Cre recombinase driven by EL222 from a non-leaky promoter[40]. We characterized the system in small-scale liquid culture (cells growing in a microfluidic chamber), larger-scale liquid cultures (batch and continuous), and short-term solid cultures (monolayer

in μIbidi slide) and found it to be functional despite changes in the context. Moreover, we established that it possesses several desirable characteristics: fast, reproducible and tunable dynamics, high efficiency for light inducible recombinases in budding yeast (Supplementary Table 2)[41–44], low leakage, and graded response of the population to light (Figs. 1 and 2). The efficiency of our system allowed us to achieve high levels of differentiation with short transient pulses that eliminate the risk of phototoxicity. Graded population responses to light were critical for achieving control of consortia composition (Fig. 3). Moreover, the high degree of reproducibility in response to light stimuli allowed us to develop a predictive model that could be used as a basis to precisely control microbial community dynamics. Using the developed model in an MPC framework allowed us to achieve bidirectional control of consortia composition in a dynamic setup (Fig. 3). We note that Klavins and colleagues[51] have developed another differentiation system in yeast, that, in principle at least, could have been used to generate consortia. This system uses a toggle switch to implement memory, and chemical inducers to toggle the switch. We believe that using light as inducer and a DNA implementation of memory allowed us to precisely characterize and select systems with appropriate properties, and drive them with the needed precision to obtain subpopulations in desired organization and proportions in space and in time, respectively.

Several solutions for the stable maintenance of microbial consortia have been proposed recently. In particular, Hasty and colleagues[52,53], Lu and colleagues[21,54], and Barnes and colleagues[17] achieved this by using synthetic biology approaches. These authors demonstrate the capacity to maintain co-cultures of several bacterial subpopulations over extended durations. However, none of these approaches succeeded at precisely controlling consortia composition. Moreover, the functioning of these systems relies on the release of signalling molecules in the environment (quorum-sensing molecules or bacteriocin) that trigger cell death. The fact that signalling molecules are released by cells creates de facto a strong dependency of the functioning on growing conditions, and notably on the density of cell cultures, an important aspect for bioproduction applications. Lastly, previous designs use elaborate genetic engineering solutions for the molecular implementation of control mechanisms, thus making extension and scaling up of these designs potentially challenging. Moreover, external control by light is inexpensive and compatible with most media composition. In summary, in comparison to previously existing solutions, our system is simple to implement, quantitatively predictable and actionable, and versatile to use.

The efficiency of the optogenetically inducible recombinase developed in this work exceeds any reported in the literature for optogenetic recombinases in yeast (Supplementary Table 2). Previous optogenetic recombination systems are based on engineering a photoactivable Cre that is typically split into two subdomains tagged with the respective photosensitive heterodimers that can be brought together upon light illumination to form a functional Cre[41–43]. However, such approaches result in activity loss as functional Cre is a tetramer and the probability of four dimerized split-Cre molecules to assemble together hinges on the relative concentrations of the two subunits[44]. A recent study reported a monogenic photoactivable Cre that is based on fusion of a LOV domain with a destabilized Cre variant[44]. The authors reported higher efficiency and stronger activation when compared to previous systems. This monogenic photoactivable Cre matched the efficiency reported for our system for up to 40 min of induction. After 40 min of induction, the activity seemed to plateau. An optogenetically inducible recombinase has been recently reported in bacteria which uses split-Cre tagged to vivid homodimers[55]. The authors demonstrate the high efficiency and

low leakage of their system at the population level. However, these properties are not quantified at the single-cell level, so a precise comparison of efficiency and leakage is not possible. Chemically inducible recombinase based systems have been employed more prominently in bacteria[56–58] and show high efficiency (>90%) but graded response or bimodal behaviour has not been reported.

Based on the principle of division of labour, microbial consortia have been employed to increase bioproduction by distributing the metabolic burden. Such approaches necessitate functional specialization in the constituent species of consortia. We provided evidence that our design can be implemented in different physiological contexts by coupling it to a growth arrest module (GAuDi system) to allow optogenetic control of growth rate and consortium composition in self-contained continuous cultures (Fig. 4). A GAuDi-like system has the potential to facilitate the switch to continuous bioproduction, touted to be the future of bioproduction[59], by separating growth and production across different subpopulations. In the context of metabolic engineering, our system could serve as a pathway switch, with the potential of compartmentalizing metabolic flux in the population. This could be achieved, for instance, by replacing fluorescent proteins by orthogonal TFs that drive entire pathways[60], leading to division of labour paradigms in consortia engineering and opening up possibilities for population level metabolic engineering.

To show that complex multispecies consortia can be created using our system, we engineered asynchronous or sequential differentiation programmes based on multiple recombination cassettes that extended the core system to generate and stably maintain multispecies consortia from a single strain in continuous liquid cultures (Fig. 5). These programmes could be scaled exponentially for applications requiring dynamic control of complex multispecies consortia and do not require intricate genetic circuits spread over multiple populations to ensure stability.

Finally, the capacity to optogenetically control cell fate decisions with spatiotemporal precision has the potential to become a critical tool for dissecting signalling pathways[24] or understanding developmental programmes[36]. Here, we showed pattern generation in 2D cultures in a microfluidic plate (Fig. 1 and Supplementary Fig. 20). Since we are not restricted to patterns attainable in nature[61], our system can provide a unique tool to study how equilibria are reached in multispecies ecosystems and elucidate microbial interactions in complex spatially structured communities.

In conclusion, we show that the system has highly desirable characteristics making it a practical tool for robustly generating and maintaining functionally distinct subpopulations both in space and in time.

## Methods

*Cloning*: All plasmids were cloned using the Golden Gate method. The majority of the used parts came from the Yeast Tool Kit (YTK)[62]. New parts, whenever necessary, were generated in the laboratory (DNA synthesis, Phusion PCR). The Golden Gate mixture was transformed via heat-shock transformation in thermocompetent *E. coli* cells and plasmids were isolated using standard miniprep kits (Macheray & Nagel, and Qiagen). Sequences of integrative plasmids and backbones can be found in Supplementary Data 1. Primers used in the study can be found in Supplementary Table 4.

*Yeast strains*: All strains used in this study are derived from BY4741 [MATa his3Δ1 leu2Δ0 met15Δ0 ura3Δ0]. Cells were transformed with linearized integrative vectors using standard lithium acetate transformation. For selection, common auxotrophic markers uracil, leucine, and histidine were used. Integrative plasmids carrying *LEU2* and *URA3* were integrated at the endogenous loci while those with *HIS3* were integrated at *HO* locus. Cells were grown in standard defined media (Sigma Aldrich Yeast Nitrogen Base) containing 2% glucose and lacking the respective auxotrophic nutrient during selection in plates (Sigma Aldrich, uracil,

leucine and, histidine drop-out media supplements). A list of strains used in this study and their genotypes can be found in Supplementary Table 5.

*Cell handling and induction*: Cells were grown overnight (ON) before the day of the experiment by picking a single colony in 50 ml Falcon tubes shaking at 200 r.p.m. at 30 °C. Following the ON, a preculture was started by diluting the ON culture 1:50. Depending on the volume of culture required, this was done in either 50 ml Falcon tubes (<15 ml) or 250 ml flat bottom Erlenmeyer flasks (<150 ml). The preculture was allowed to grow for at least 3 h (~2 cell generations). Care was taken to do all the experiments in the exponential phase. Induction was done by using RGB led strips (Adafruit NeoPixel Digital RGB LED Strip). Max LED intensity was set at 40 for all experiments unless specified (max led intensity possible is 255). The control was performed via Arduino microcontrollers (Genuino Uno and Mega). Cells were grown in synthetic complete low fluorescence media with 2% glucose for all experiments. Light-sensitive strains were grown in the dark. All manipulations were performed in the presence of red light (Supplementary Note 8).

*Batch culture*: Cells were grown to the exponential phase in the dark from ON culture in Falcon tubes shaking at 200 r.p.m. at 30 °C in a custom Falcon tube before starting the experiment. Induction was carried out in a custom Falcon tube holder fitted with LED strips.

*Turbidostat*: Turbidostat refers to our custom LED-equipped continuous culture platform and control software, ReacSight (Supplementary Note 1 and Supplementary Fig. 1)[28]. Experiments done in the turbidostat followed a similar protocol as described above (see section on cell handing). Cells were allowed to grow in the dark until exponential phase and induction was started only after the growth rate stabilized. Samples were taken automatically from the turbidostat at regular intervals, diluted 20 times with a pipetting robot and passed through the cytometer. The entire vessel, including pumps and tubing, was autoclaved before each experiment. Unless stated differently, the experiments used a "grow and dilute" programme where cells were allowed to grow until OD 0.6 and then diluted to OD 0.4. The growth rate was computed by calculating the slope of linear curve fit to the log of OD data with time. Information regarding the times when the dilutions took place was stored, in addition to the OD and LED status as csv files. Data were acquired and preprocessed using the ReacSight software framework. Subsequent treatment of the data was done in Python.

*Cytometry and data analysis*: All cytometry measurements were acquired with a Guava EasyCyte BGV 14HT benchtop cytometer using the InCyte software (version 3.3). The settings were kept constant for all experiments. For turbidostat experiments 5000 events were recorded for each sample, except for leakage and efficiency experiments (Fig. 1c), for which, 50,000 events were recorded. No compensation was used during acquisition. Data were deconvolved after acquisition (Supplementary Figs. 4 and 5). Data were gated using kernel density-based methods (Supplementary Fig. 3). Python was used for data analysis and visualization (Supplementary Note 1).

*Microscopy*: Microscopy was performed on the inverted microscopy platform Leica DMi8 S. Live cell imaging was either performed in Ibidi μ-slide VI 0.4 (80606) or CellASIC ONIX platform with the microfluidic plates (Y04C) provided by the vendors. The details for the exact excitation and emission spectra of the fluorophores and filters used can be found in Supplementary Table 1. Unless specified differently, the interframe interval was 6 min. During time-lapse live cell imaging, the chamber temperature was maintained at 30 °C. We used an in-house software, called MicroMator, for the automated acquisition and cell tracking. Cell segmentation was achieved via SegMator, an in-house neural net-based segmentation algorithm[30]. Python was used for data analysis and visualization (Supplementary Note 2). FIJI (ImageJ 1.52i) was used for the processing of images shown in the manuscript.

*ODE model fitting and parameter estimation*: ODE models were solved using solve_ivp solver from SciPy.integrate library. Models were fitted using the least-square method from SciPy.optimize library. Parameter search for each model was done locally (gradient descent) with multiple initial guesses for parameters and bounds on parameter values between $10^{-10}$ and 10. To account for the delays, mean squared deviations were calculated after shifting model predictions by 60 min for the differentiation system ODE model and 120 min (differentiated & non-differentiated), 360 min (dead), and 300 min (growth rate) for the GAuDi ODE model.

*MPC experiments*: ODE models were used in an MPC framework[28]. The framework consisted of solving the model given a light sequence. This sequence was then optimized using a least-square method from SciPy.optimize library to minimize the error between predictions and the target set point starting from an initial state. The optimization for light sequence was done for a time horizon of 5 h in the form of 10 duty cycles of 30-min period. Cultures were sampled every hour for two reactor control experiments (Fig. 3b, c) and every 2 h for single reactor control experiments (Fig. 3e). The light sequence was updated at each timepoint. To estimate the state of the system, cytometry data were analysed online to determine the state at the time of sampling. This estimate of sampling time state was then used as initial conditions for the model and the current state was estimated by solving the model for a time, τ. τ stands for a delay, consisting of a sampling delay (time passed between sample acquisition and finalized data analysis) and an observation delay (time required for enough fluorescent protein to accumulate to pass the differentiation threshold) (Supplementary Note 5).

**Reporting summary**. Further information on research design is available in the Nature Research Reporting Summary linked to this article.

## Data availability
Raw cytometry and OD data generated in this study and processed microscopy data have been deposited on Zenodo under the accession code 4923833. Plasmids and strains are available from the corresponding authors upon request.

## Code availability
Jupyter notebooks used for data analysis are available online in the GitLab repository (https://gitlab.inria.fr/InBio/Public/YeastOptogeneticDifferentiation_YODA). The notebooks also include code for model simulation and fitting.

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

## Acknowledgements

We thank Sebastian Sosa Carrillo for providing plasmid backbones, Steven Fletcher and Achille Fraisse for assistance with the microscope acquisition software, Virgile Andreani for help with model fitting, and Zachary Fox for providing software for image analysis. We thank the members of the Batt group, Gael Yvert, and Hyun Youk for conducive discussions. We are grateful to Mustafa Khammash (ETH Zurich), Bernd Mueller-Roeber (University of Potsdam), and Matthias Peter (ETH Zurich) for providing plasmids. C.A. is enrolled in the Frontières de l'Innovation en Recherche et Education (FIRE) doctoral school hosted by Université de Paris. This work was supported by the H2020 Fet-Open COSY-BIO grant (grant agreement no. 766840), by the Inria grant IPL COSY and by ANR grants CyberCircuits (ANR-18-CE91-0002), MEMIP (ANR-16-CE33-0018), and Cogex (ANR-16-CE12-0025).

## Author contributions

C.A. conceived and cloned the biological circuits and performed experiments. C.A. and F.B. did mathematical modelling and analysed data. C.A., J.R. and G.B. wrote the paper with inputs from F.B. J.R. and G.B conceived and supervised the study.

## Competing interests

The authors declare no competing interests.
