## [Peer Review File · Nature Communications]

Reviewers' Comments:

Reviewer #1:

Remarks to the Author:

This manuscript describes a light-inducible system for permanently differentiating yeast cells via a Cre recombinase. By linking the system to feeder turbidostats and growth rate limiters they show that a model guided computer controller can maintain and desired ratios of cells types.

Overall, I think this is a really fantastic paper that should be of interest to a wide variety of scientists. The results are pretty clear cut and I believe the experiments were done well with the proper controls.

My only minor quibbles are:

1. The authors (on line 99) state that the efficiency of their recombinase system is unprecedented. I think this is a bold claim, and there is no evidence to back up the assertion.
2. There have been a couple other novel differentiation systems recently (in *E. coli*) that should probably be mentioned or discussed.

Reviewer #2:

Remarks to the Author:

The creation of synthetic microbial consortia is of great interest for industrial biotechnologists, synthetic biologists and microbiologists. The authors describe the development of yeast strains which can be differentiated into distinct sub-populations by the application of blue light. This system relies on the previously described EL222 blue light responsive transcription factor and the Cre/Lox recombination system.

A simple mathematical model is developed and the behaviour of this strain is thoroughly characterised. This enables the use of model predictive control to deliver blue light input to maintain continuous cultures at target population ratios. A growth arrest system was included so that the culture did not have to be continuously renewed with non-differentiated cells. Finally, the cells were modified with a multi-fate differentiation system, with both sequential and asynchronous paths demonstrated.

This is a wonderful piece of work which blends the worlds of synthetic biology and control in a well thought of narrative. The work in this manuscript was thorough, well described and the authors were honest with the limitations of their system.

Major Comments:

One of the advantages of this system over previous community control is that community composition and population density can be controlled. Or as stated in the discussion, this system is not dependent on the density of cell cultures. However, from what I can see, all the experiments are performed with the same target OD. Although I don't believe that it is necessary for publication, the story feels incomplete without demonstration of maintaining target composition at different densities. Does the fact that, in simplifying the mathematical model, the strains' growth rates have been fixed based on measurements at a particular OD, complicate this?

General Minor Comments:

The efficacy of the EL222 + Cre system is impressive. Does your data concur with previous uses of EL222 or does it exceed prior work? Do you have an understanding as to why EL222 is so superior to other light inducible systems in yeast?

I think it is excellent that the authors have tested the function of the EL222 + Cre system in multiple contexts. This is incredibly rare in this type of work and I congratulate the authors. However, the spatial control feels as though it is bolted on rather than an integral part of this work. I think the authors could emphasise that they have shown the differentiation system

functioning small scale liquid cultures (microfluidic), larger scale liquid cultures (bioreactors), AND solid cultures.

The design of the sequential differentiation system is not intuitive. The Cre and EL222 genes are included within the mNeon excision cassette. Is this simply to add length to the cassette? If so, it seems an odd choice to include genes which are fundamental to the behaviour of the system. If not, can you include explanation for why these parts need to be removed? Further, this system still includes the FAR1M system but not the positive feedback. Does this mean that you do not see growth arrest of differentiated cells and as such you don't get replenishment of undifferentiated population?

In the two reactor setup, you state that the densities of the two reactors are different. How are they maintained at different densities? If the flow from the first is in to the second, then the dilution rates will be the same? Is there some flow from the first that is diverted? Is there some sterile media that is fed into the second?

Specific Minor Comments:

Line 96: How was the threshold determined? Does the mCerulean measurement mirror the mNeon measurement?

Lines 105 and 107: I suggest avoiding the use of "frequency" here as it may confuse readers into believing you are varying the wavelength of the light.

Line 121: How was this light profile ("1s every 3 minutes for an hour") decided?

Line 122: Why were cells kept in darkness for an hour? To allow fluorescence maturation?

Line 125: I am not sure why this combination indicates prior differentiation. This seems like the profile you would expect from any differentiated cell.

Lines 141-142 and Fig 2a: This diagram is slightly hard to understand. Are there only two 30 minute pulses at x minutes apart, or are there several 30 minute pulses x minutes apart?

"Repeated" implies several but the image indicates two.

Line 148-149: There is a small offset from $y=x$. Is there an explanation for this or is it assumed to be within some measurement error?

Line 152-153: Why fit to two data sets separately?

Line 181-182: The theoretical limit of differentiated cells that you could reach was 90%. Did you try this? I assume it would just be a constant illumination signal.

Line 188: If the data is available it would be good to show this deviation from the target once control was stopped.

Fig S4.3: The response becomes dampened after repeated cycles. Is this due to accumulation of escaped cells?

Lines 211-212: You might want to state that escape from any engineered growth arrest/killing system is a common occurrence. See:

M. O. Din et al., "Synchronized cycles of bacterial lysis for in vivo delivery," *Nature*, vol. 536, no. 7614, pp. 81–85, Aug. 2016.

F. K. Balagaddé et al., "A synthetic Escherichia coli predator-prey ecosystem," *Mol. Syst. Biol.*, vol. 4, no. 1, p. 187, Jan. 2008.

Line 220: "non-trivial light signal". Is this a random signal, a signal that has been determined by some form of DOE to produce the best data for fitting, etc. ...?

Lines 223-224: Can you point to a specific example? Fig S4.4b?

Line 330: The sequences are included as unannotated DNA strings. For the sake of researchers trying to understand your plasmids, I would encourage you to include annotated DNA sequences.

Fig 1a: Terminators are fundamental to this system so I think they should be included in the cartoon. It is not intuitive that mNeon is not expressed in the undifferentiated state.

Fig 1b: Are the cells with intermediate brightness due to slow maturation of the mNeon?

Fig 3b: Is the deviation of the consortium at 60% towards the end of the experiment due to escape?

Fig 5b: The double differentiated strain looks like it is being outgrown by the single differentiated strains. Is this expected? Also, after a single 30 minute pulse, we see lots of double differentiated cells. Have you managed to capture a situation in which you have single differentiated but no double differentiated cells?

Response to reviewers

We thank both reviewers for their positive comments of our work and for their suggestions to improve the article.

Below we provide detailed comments to their remarks. In what follows, reviewers' comments are in **black**, response to reviewers are in **blue** and the modified text is in **green** with changes **highlighted**.

Reviewer #1:

This manuscript describes a light-inducible system for permanently differentiating yeast cells via a Cre recombinase. By linking the system to feeder turbidostats and growth rate limiters they show that a model guided computer controller can maintain and desired ratios of cells types.

Overall, I think this is a really fantastic paper that should be of interest to a wide variety of scientists. The results are pretty clear cut and I believe the experiments were done well with the proper controls.

We thank reviewer #1 for their positive assessment of our manuscript.

My only minor quibbles are:

1. The authors (on line 99) state that the efficiency of their recombinase system is unprecedented. I think this is a bold claim, and there is no evidence to back up the assertion.

We agree that this is a strong claim but note that we only claim unprecedented efficiency for optogenetic recombination systems in yeast. In response to the reviewer's comment, we have reformulated the sentence to better clarify that we refer only to optogenetic recombinases in yeast and that, for yeast, the claim is backed up by Table S.2 in the supporting information.

Line 67. We demonstrate that our system shows desirable features including low background activity, unprecedented efficiency for optogenetic recombinases in budding yeast, graded response to varying light signals, absence of hysteresis, and dynamics that are fast, predictable, and tunable.

Line 99. Moreover, the efficiency of the system in budding yeast was unprecedented, when compared to existing light inducible systems, leading to >99 % differentiation after 4h of induction (Figure 1c, blue inset) (Table S2)¹⁻⁴.

Line 266. We characterized the system and established that it possesses several desirable characteristics: fast, reproducible and tunable dynamics, unprecedented efficiency for light inducible recombinases in

budding yeast (Table S2) 42–45, low leakage, and graded response of the population to light (Figure 1 & 2).

Publication	System	Growth conditions	Efficiency (mean ± s.d.) - Induction time	Leakage in dark (mean ± s.d.)
Ref 6 (Figure 3), adapted from Ref 4 [†]	split Cre CIB1_CRY2	Stationary liquid culture	1.6% ± 0.8 at 90 mins	1.3% ± 0.5 over 24h of dark culture
Ref 6(Figure 3), adapted from Ref 5 ^{††}	split Cre pMag-nMag	Stationary liquid culture	21.2% ± 5.8 at 90 mins	7.1% ± 1.1 over 24h of dark culture
Ref 7 (Figure 2) original study	split Cre PhyB-PIF3*	Stationary liquid culture	46.7% ± 5.3 at 24 h**	6.7% ± 2.6 over 24h of dark culture
Ref 6 (Figure 3 & 4) original study	destabilized Cre fused to asLOV2 (LiCre)	Stationary liquid culture Exponential liquid culture	41.2% ± 2.8 at 40 mins 66.7% ± 3.7 at 90 mins 66.8% ± 3.3 at 180 mins 7.6% ± 2.1 at 40 mins	0.7% ± 0.2 over 24h of dark culture
This study	EL222 inducible WT Cre	Exponential liquid culture	43.1% ± 2.7 at 40 mins 76.8% ± 1.7 at 90 mins 94.4% ± 0.9 at 180 mins 99.7% ± 0.1 at 240 mins	0.06% ± 0.05 over 72h of dark culture

2. There have been a couple other novel differentiation systems recently (in E. coli) that should probably be mentioned or discussed.

We are not sure which papers exactly the reviewer has in mind. Several recombinase based systems^{6–8} have been reported in bacteria but only one study has reported using light as an inducer⁹.

In any case, thanks to the reviewer’s comment, we have added a section on optogenetic recombinases, including Sheets et al., in the discussion and we would be happy to include additional discussions if the reviewer can point us to more references.

Line 297. “The efficiency of the optogenetically inducible recombinase developed in this work exceeds any reported in the literature for optogenetic recombinases in yeast (Table S2). Previous optogenetic recombination systems are based on engineering a photoactivable Cre that is typically split into two subdomains tagged with the respective photosensitive heterodimers that can be brought together upon light illumination to form a functional Cre^{1,2,5}. However, such approaches result in activity loss as functional Cre is a tetramer and the probability of four dimerized split-Cre molecules to assemble together hinges on the relative concentrations of the two subunits⁴. A recent study reported a monogenic photoactivable Cre that is based on fusion of a LOV domain with a destabilized Cre variant⁴.

The authors reported higher efficiency and stronger activation when compared to previous systems. Indeed, this monogenic photoactivable Cre matched the efficiency reported for our system for up to 40 minutes of induction. After 40 minutes of induction, the activity seemed to plateau for the former. A novel optogenetically inducible recombinase has been recently reported in bacteria which uses split Cre tagged to vivid homodimers⁹. The authors demonstrate the high efficiency and low leakage of their system at the population level. However, these properties are not quantified at the single cell level, so a precise comparison of efficiency and leakage is not possible. Chemically inducible recombinase based systems have been employed more prominently in bacteria⁶⁻⁸ and show great efficiency (>90%) but graded response or bimodal behaviour has not been reported.

Reviewer #2 (Remarks to the Author):

The creation of synthetic microbial consortia is of great interest for industrial biotechnologists, synthetic biologists and microbiologists. The authors describe the development of yeast strains which can be differentiated into distinct sub-populations by the application of blue light. This system relies on the previously described EL222 blue light responsive transcription factor and the Cre/Lox recombination system.

A simple mathematical model is developed and the behaviour of this strain is thoroughly characterised. This enables the use of model predictive control to deliver blue light input to maintain continuous cultures at target population ratios. A growth arrest system was included so that the culture did not have to be continuously renewed with non-differentiated cells. Finally, the cells were modified with a multi-fate differentiation system, with both sequential and asynchronous paths demonstrated.

This is a wonderful piece of work which blends the worlds of synthetic biology and control in a well thought of narrative. The work in this manuscript was thorough, well described and the authors were honest with the limitations of their system.

We thank reviewer #2 for their positive summary of our manuscript.

Major Comments:

One of the advantages of this system over previous community control is that community composition and population density can be controlled. Or as stated in the discussion, this system is not dependent on the density of cell cultures. However, from what I can see, all the experiments are performed with the same target OD. Although I don't believe that it is necessary for publication, the story feels incomplete without demonstration of maintaining target composition at different densities.

We agree that demonstrating functionality of our system at different optical densities would strengthen the manuscript. We have therefore performed experiments in which the cell density is increased up to 15-fold and cells are exposed to multiple 30min light pulses (with sufficient time between subsequent light pulses to allow for quantification of the recombined fraction per pulse). We find that the functioning of our differentiation system is not significantly affected by the increased cell density. These new results are now included in the supporting information to back up the claim that our system can in principle be used at different ODs.

End of Section IIIa

Next, we asked the question whether the functioning of the system remains predictable in light of 15 fold variation in cell density. We found that the cell density does not significantly affect the differentiation dynamics strongly suggesting that the system can be used for dynamic control purposes at different ODs.

Figure S3.5. a. Comparing the variability in response to the same (30-min) pulse when cells are cultured continuously at different ODs. For each OD, a single experiment was performed in which cells have been exposed to five 30min light pulses delivered 3h apart from each other and the differentiated fraction in response to each pulse was quantified (diamonds). b. Dynamics of differentiation at different ODs in response to repeated pulses of 30 mins that were used to quantify differentiation per pulse in a. Colors in b correspond to the ODs in a (see x-ticks).

Does the fact that, in simplifying the mathematical model, the strains' growth rates have been fixed based on measurements at a particular OD, complicate this?

The reviewer is right in assuming that the controller model would have to be updated with the growth rates at the new cell densities for the control target to be reached. To demonstrate that it is possible to control the system at different ODs, we share data from a preliminary MPC experiment done with another strain. In this experiment, both the reservoir and the control reactor were maintained at an OD of 0.5. We did not include this experiment in the study due to hardware malfunctions (not related in any way to the target cell density of the cultures) and because this experiment was done with a preliminary strain/version of our system that is not included in the paper.

Figure R1. Differentiation system in a different strain from the one reported in the study. The red dashed line shows the desired target and circles represents the differentiation fraction. Control was started at $t=31.83h$ and at an OD of 0.5.

General Minor Comments:

The efficacy of the EL222 + Cre system is impressive. Does your data concur with previous uses of EL222 or does it exceed prior work? Do you have an understanding as to why EL222 is so superior to other light inducible systems in yeast?

We were able to reproduce the results of Benzinger and Khammash, 2018 in terms of expression levels and modulation of noise in expression. One point of difference between our study and previous studies is the choice of constitutive promoter for expressing EL222. Typically, use of pTDH3 is disfavored due to dependency on glucose concentration but because we operate in continuous phase this is not an issue. This allowed us to obtain almost double EL222 levels compared to previous studies¹⁰⁻¹², which use pACT1 and pPGK1 both of which are around half as strong as pTDH3; Therefore, perhaps in our system we have higher expression of the target gene. We believe that for our application, the EL222 system was superior to other systems reported in yeast because

- It affords tighter control over background activity in the dark (non-leakiness)

- It permits to tune the strength and/or the variability in expression
- It is a homodimer (therefore requires cloning of a single gene) and does not rely on the addition of expensive chromophores

We now detail the reasons for choosing the EL222 system in the supplementary information.

First line of Section IIIa.

While several solutions exist for optogenetic expression in yeast [^{11,13–15}] we decided to use EL222 to drive Cre because it possessed several desirable features like,

- **tighter control over background activity in the dark (range of promoters)**
- **control over the strength and/or the variability in expression**
- **it is a homodimer (therefore requires cloning of a single gene) and does not rely on the addition of expensive chromophores**

I think it is excellent that the authors have tested the function of the EL222 + Cre system in multiple contexts. This is incredibly rare in this type of work and I congratulate the authors. However, the spatial control feels as though it is bolted on rather than an integral part of this work. I think the authors could emphasise that they have shown the differentiation system functioning small scale liquid cultures (microfluidic), larger scale liquid cultures (bioreactors), AND solid cultures.

The comment is well received. We have added sentences in the discussion to emphasise the spatial control. We thank the reviewer for highlighting this contribution.

Line 266. We characterized the system **in small scale liquid culture (cells growing in a microfluidic chamber), larger scale liquid cultures (batch and continuous), and short-term solid cultures (monolayer in μ lbidi slide) and found it to be functional despite changes in the context.**

Line 278. We believe that using light as inducer and a DNA implementation of memory allowed us to precisely characterize and select systems with appropriate properties, and drive them with the needed precision to obtain subpopulations in desired **organization and proportions in space and in time, respectively.**

The design of the sequential differentiation system is not intuitive. The Cre and EL222 genes are included within the mNeon excision cassette. Is this simply to add length to the cassette? If so, it seems an odd choice to include genes which are fundamental to the behaviour of the system. If not, can you include explanation for why these parts need to be removed? Further, this system still includes the FAR1M system but not the positive feedback. Does this mean that you do not see growth arrest of differentiated cells and as such you don't get replenishment of undifferentiated population?

Our intention in removing Cre and EL222 upon excision of the longer site was to lock cells into the singly recombined state if they recombine the longer site before the short one. However, the fraction of cells that recombined the longer site first was too low for this feature of the design to matter much.

The inclusion of FAR1M without the positive feedback loop on ATAF1 does not induce a growth arrest, however, it does introduce a slight growth defect. The authors believe that over longer time scales the undifferentiated and single recombined (at C site) species can be replenished.

Figure R2. A strain containing only the N recombination cassette and expressing FAR1M without the feedback loop was cultured continuously and maintained above 80% differentiation. Induction was started at $t=9h$. The differentiation fraction decreased slowly after initial differentiation suggesting slight difference in growth.

In response to the reviewer's comment, clarifications on both these points are now included in the supplementary material Section: VII *Expanding the differentiation system to give rise to multi-species consortia with differentiation programs.*

Third paragraph of the section VII,

Note that EL222 and Cre are included within the flowed region and, therefore, following recombination, are removed from the cell. Our intention in removing Cre and EL222 upon excision of the longer site was to lock cells into the singly recombined state if they recombine the longer site before the short one. However, the fraction of cells that recombined the longer site first was too low for this feature of the design to matter much.

We also note that the expression of FAR1M does have an effect on the growth rate of \overline{CN} and $C\overline{N}$, however, the difference was not large enough to be observable at the timescale of the induction profiles used in the experiments shown in the study (Figure 5b & d). The authors believe that it will be possible to replenish the CN and \overline{CN} species over longer timescales in the dark.

In the two reactor setup, you state that the densities of the two reactors are different. How are they maintained at different densities? If the flow from the first is in to the second, then the dilution rates will be the same? Is there some flow from the first that is diverted? Is there some sterile media that is fed into the second?

The control logic to maintain the OD of the two connected reactors at different levels is indeed quite subtle. Due to the inflow from the reservoir culture, maintained at a lower cell density, the control culture required less fresh media to maintain the target cell density but fresh media was fed to the control reactor whenever the OD was above the set target.

More specifically, the cultures are maintained at target cell densities in fixed volumes by continuously measuring the OD every 5 minutes and adding approximately 1ml of media (opening the input pump for 6s (pump flow rate ~10-15 ml/min)) if the OD exceeds the target. This is followed by opening the output pump which drains the culture if it is above a certain level to ensure that the volume does not increase. Since the OD control for each vessel is independent, this can be achieved relatively easily. It becomes complicated, as the reviewer guessed, when the output of the reservoir is connected to the control reactor. In this scenario, the dilution rate of the reservoir reactor is its growth rate (since it is maintained at a constant cell density in a fixed volume). All the outflow from the reservoir reactor is fed to the control reactor and this leads to two things, an increase in the volume of the control culture and its dilution. The former was addressed by increasing the duration for which the outflow pump is opened at each measurement for the control reactor.

Thanks to the reviewer's comment we have clarified this point in the main text.

Line 167. **In addition to the feed from the reservoir reactor, the control reactor was fed fresh media to maintain the culture at the target cell density.**

Specific Minor Comments:

Line 96: How was the threshold determined?

The threshold was determined by observing the evolution of GRN-B-HLin (mNeonGreen) fluorescence over time and a threshold of 200 was set such that only cells that have expressed detectable amounts of mNeonGreen are considered differentiated.

We now provide this information in the main text in the caption to Figure 1b.

Line 554. **The threshold was set such that only cells that have expressed detectable amounts of mNeonGreen are classified as differentiated.**

Does the mCerulean measurement mirror the mNeon measurement?

The mCerulean fluorescence does mirror mNeonGreen fluorescence. However, due to high autofluorescence in the channel used to detect mCerulean in raw cytometry data and low brightness of mCerulean, we do not observe two well separated peaks for the differentiated and the non-differentiated population. Using deconvolution allows us to see a clearer separation between the differentiated and non-differentiated populations (Figure R3).

Figure R3. Histograms of total mCerulean and mNeonGreen fluorescence in the population over time. Induction was started at $t = 8.5h$. Up to $8.5h$ (red distributions) cultures consist of a predominantly non-differentiated population. From $10h$ - $15h$ (yellow distributions) cells have recombined but yet not completely lost all the mCerulean protein and $16h$ (blue) onwards cultures contain predominantly differentiated cells.

Note that in the paper, raw fluorescence has been used to ascertain differentiation (Figure 1b).

Lines 105 and 107: I suggest avoiding the use of “frequency” here as it may confuse readers into believing you are varying the wavelength of the light.

We thank the reviewer for pointing this out and we have changed all instances of “frequency” by “duration”.

Line 105 & Line 107 frequency to duration

Line 107 We note that the differentiated fractions are reminiscent of EL222 inducible fluorescent protein levels obtained when varying the light intensity or the frequency **or duration** of applied light pulses.

Line 121: How was this light profile (“1s every 3 minutes for an hour”) decided?

The light profile was chosen based on the characterization experiments in the microfluidic plate where induction was carried out by delivering 1s of light every 6 mins (Figure 1d). The Ibidi slide could not support a monolayer of yeast cells for more than 2.5-3h. Eventually bubbles would emerge disturbing the monolayer and ruining the experiment. It took about 30 minutes for all the floating cells to settle. We found that cells required at least 1h to generate enough mNeonGreen to be readily detectable (Supplementary Figure 2.2d). This detection issue was exacerbated by the 10X objective we used in these experiments. Due to these constraints, 1h was the maximum induction that led to reproducible patterns. Increasing beyond this induction period did not result in more detectable differentiation in the timeframe of the experiment. 1s of light every 6 minutes also led to differentiation within the pattern but the efficiency (qualitatively) was lower. By doubling the number of pulses, we could achieve better efficiency (qualitative). This is consistent with characterization in the microfluidic device (Figure 1d).

In response to the reviewer’s query, we have expanded the discussion on pattern formation experiments in the supplementary information section VI: Spatial control and the main text refers to this discussion.

Line 121. Cells were illuminated with a given pattern for 1s every 3 minutes during 1h (Supplementary information, section VI).

Supplementary information, section VI

Based on the results of the characterization experiment, we initially chose to induce the cells with 1s pulses every 6 mins at an intensity of 10%. The ibidi slide could not support a monolayer of yeast cells for more than 2.5-3h. Eventually bubbles would emerge disturbing the monolayer. It took about 30 minutes for all the floating cells to settle. We found that cells required at least 1h to generate enough mNeonGreen to be readily detectable (Supplementary Figure 2.2d). Due to these constraints, 1h was the maximum induction that led to reproducible patterns. Increasing beyond this induction period did not result in more detectable differentiation in the timeframe of the experiment. To improve the differentiation efficiency, we finally decided to double the frequency of pulses.

Line 122: Why were cells kept in darkness for an hour? To allow fluorescence maturation?

Indeed, the reason for the relaxation was to allow enough mNeonGreen molecules to be expressed and mature. Also to let the cells recover after being stressed by the high intensity light. The former is now explicitly stated in the main text.

Line 121. Following this, cells were kept in darkness for an hour prior to imaging in order to ensure **a good assessment of the differentiation state of cells (time for the mNeonGreen protein to be produced and matured).**

Line 125: I am not sure why this combination indicates prior differentiation. This seems like the profile you would expect from any differentiated cell.

It indicates prior differentiation because the time of the experiment does not allow the mCerulean fluorescence to be lost via dilution. For the same reason, mNeonGreen fluorescence should not be at maximum levels. The fact that there is no mCerulean along with high mNeonGreen fluorescence strongly suggests that these cells recombined long time before the experiment. See response to first specific minor comment.

To address the reviewer's comment, we have clarified this point in the main text.

Line 123. Some recombination was present outside of the provided pattern **but it is very likely that these cells had differentiated long before the start of the experiment since they lack mCerulean fluorescence.**

Lines 141-142 and Fig 2a: This diagram is slightly hard to understand. Are there only two 30 minute pulses at x minutes apart, or are there several 30 minute pulses x minutes apart? "Repeated" implies several but the image indicates two.

Indeed. We agree that it was a bit ambiguous, and we have now mentioned explicitly in the caption of the figure that this diagram is representative and pulses were repeated at the said intervals for the duration of the experiment.

Line 567. Induction was in the form of either single pulses of different durations (top) or repeated pulses of 30 minutes at different interpulse intervals (bottom; **only the first two pulses are represented**).

Line 148-149: There is a small offset from $y=x$. Is there an explanation for this or is it assumed to be within some measurement error?

This is a very good observation. We cannot say for certain as this could be explained by the reactor to reactor variability in the system and/or the measurement error. However, the systemic nature of the deviation suggests that there might be a slight non-linearity in the differentiation response of the system w.r.t. pulse duration. We now clarify this point in the main text.

Line 148. We found that continuous light resulted in **similar** differentiation fractions as discrete pulses for the same total duration of induction (Figure 2f).

Line 152-153: Why fit to two data sets separately?

The two datasets were fit separately to highlight the predictable behaviour of the system. In other words, to show that different datasets lead to the same estimate of the differentiation rate.

Line 181-182: The theoretical limit of differentiated cells that you could reach was 90%. Did you try this? I assume it would just be a constant illumination signal.

We did not try reaching 90% differentiation for the set of experiments shown in the paper although in a preliminary experiment we observed that for a theoretical maximum target of 80%, the control signal was continuous light.

Line 188: If the data is available it would be good to show this deviation from the target once control was stopped.

We do not have data to show deviation from the control when the light was intentionally stopped for the experiment sets shown in the paper. However, we share data from an earlier experiment with a different controller that led to a set point error (because we used a too high differentiation rate in the model). Control was lost at the end of this experiment (last 5 time points) due to a hardware malfunction. We do not include this figure in the manuscript because this experiment used an old version of the controller that led to a set-point error.

Figure R4. Loss of control in absence of light signal for two-vessel control. This control experiment was performed with a different controller and showed a set point error with respect to the target. The red dashed line shows the desired target, which followed a step-like profile. Induction was started at $t=7h$. Circles represent the differentiation fraction. Due to equipment malfunction light could not be delivered after 63h. Black triangle denotes deviation from the target once the control was stopped.

Fig S4.3: The response becomes dampened after repeated cycles. Is this due to accumulation of escaped cells?

Accumulation of escaped cells is one possible explanation of this observation. However, the escapers we could isolate (not from this particular experiment but from other experiments) continued to express mScarlet-I. It is therefore more likely that the dampening is due to accumulation of dead cells towards the end of the experiment. The light pulses are delivered at a decreasing interpulse period (17h between the first and the second to 9h between the penultimate and the last one). The low interpulse duration leads to accumulation of dead cells. This is also predicted by the model (indicated by black arrows).

Figure R5. Experiment from Figure 4c. Black arrows indicate the accumulation of dead cells with shorter interpulse duration and a lower decrease in the growth rate compared to previous pulses.

Following the comment of the reviewer, we have included a sentence in the supplementary text concerning the dampened response in the caption of Figure S4.4.

We note that in c., light pulses were applied with a decreasing interpulse period and that the response became dampened as the interpulse period decreased. Data suggests the cause to be accumulation of dead cells. Model predictions are in agreement with this hypothesis.

Lines 211-212: You might want to state that escape from any engineered growth arrest/killing system is a common occurrence. See:

M. O. Din et al., "Synchronized cycles of bacterial lysis for in vivo delivery," *Nature*, vol. 536, no. 7614, pp. 81–85, Aug. 2016.

F. K. Balagaddé et al., "A synthetic *Escherichia coli* predator–prey ecosystem," *Mol. Syst. Biol.*, vol. 4, no. 1, p. 187, Jan. 2008.

We thank the reviewer for highlighting this aspect and a sentence to address this point is now included in the main text.

Line 233. Evolutionary constraints are known to limit the long-term stability of synthetic circuits¹⁶ particularly circuits that are engineered to implement growth arrest / self-killing at the population level^{17,18}.

Line 220: "non-trivial light signal". Is this a random signal, a signal that has been determined by some form of DOE to produce the best data for fitting, etc. ...?

We did not use optimal experimental design to choose the light signal. We chose a light signal comprised of various pulses of different duration at different interpulse periods because the intuitive expectation is that such an experiment would lead to data that is informative for learning model parameters.

To address the reviewer's comment, we have supplemented "non-trivial" by "pulses of varying durations at different intervals".

Line 219. The model was fitted to dynamical data from an experiment with a non-trivial light signal **comprised of pulses of varying durations at different intervals** (Figure 4b, Supplementary text IV).

Lines 223-224: Can you point to a specific example? Fig S4.4b?

The comment of the reviewer is well received, and the suggested modification has been made in the main text. The authors would like to point out that the agreement with the data was observed for each experiment shown in S4.4.

Line 223. Figure S4.4b

Line 330: The sequences are included as unannotated DNA strings. For the sake of researchers trying to understand your plasmids, I would encourage you to include annotated DNA sequences.

The comment of the reviewer is well received, and the annotated plasmid sequences are included as genbank files in the supplementary information. Note that we also corrected the internal ID of two plasmids that have been mistakenly exchanged in the submitted supplementary files.

Fig 1a: Terminators are fundamental to this system so I think they should be included in the cartoon. It is not intuitive that mNeon is not expressed in the undifferentiated state.

The comment of the reviewer is well received, and terminators have now been added to the cartoons in the main text.

Fig 1b: Are the cells with intermediate brightness due to slow maturation of the mNeon?

We believe that it is hard to say whether it is actually due to the slow maturation of mNeonGreen or due to the time required to generate enough mNeonGreen to be detectable. Microscopy data seems to suggest the latter but it is not obvious.

Fig 3b: Is the deviation of the consortium at 60% towards the end of the experiment due to escape?

We do not believe that this is because of escape. It is hard to say with absolute certainty but it is possible that the feed from the reservoir was diminished towards the end of the experiment probably due to a faulty pump.

Fig 5b: The double differentiated strain is looks like it is being outgrown by the single differentiated strains. Is this expected?

The reviewer's keen eye is appreciated. As mentioned in an earlier response, the expression of FAR1M does not induce a growth arrest but decreases the growth rate slightly. So over time we do expect the double recombined cells and N recombined cells to be outgrown by unrecombined cells and C recombined cells. However, the timescale of this experiment does not permit us to observe this. Instead, cells that are singly recombined at the N locus possess both mNeonGreen and mScarlet-I for some time. Once

mNeonGreen is diluted out, the cells are left with just mScarlet-I and therefore are classified as singly recombined. In short, the decrease we see is an artefact of the threshold based quantification that is limited by the dilution of mNeonGreen.

This is now clarified in the main text in the caption of Figure 5b.

Line 640. **We note that the small decrease in the fraction of \overline{CN} cells just after differentiation and a corresponding increase in CN is an artefact of the threshold-based quantification that is limited by the dilution of mNeonGreen.**

Also, after a single 30 minute pulse, we see lots of double differentiated cells. Have you managed to capture a situation in which you have single differentiated but no double differentiated cells?

We have not observed a situation where we have only the single differentiated cells but no double differentiated cells in any experiment performed so far. It is an interesting possibility. We expect that lower intensities or short pulses under the microscope might result in such events. Alternatively, increasing the length of the to-be-excised-fraction should yield results where the single recombined species are present at a higher prevalence than the double recombined.

References

1. Taslimi, A. *et al.* Optimized second-generation CRY2-CIB dimerizers and photoactivatable Cre recombinase. *Nat. Chem. Biol.* **12**, 425–430 (2016).
2. Kawano, F., Okazaki, R., Yazawa, M. & Sato, M. A photoactivatable Cre-loxP recombination system for optogenetic genome engineering. *Nat. Chem. Biol.* **12**, 1059–1064 (2016).
3. Hochrein, L., Mitchell, L. A., Schulz, K., Messerschmidt, K. & Mueller-Roeber, B. L-SCRaMble as a tool for light-controlled Cre-mediated recombination in yeast. *Nat. Commun.* **9**, 1–10 (2018).
4. Duplus-Bottin, H. *et al.* A single-chain and fast-responding light-inducible Cre recombinase as a novel optogenetic switch. *Elife* **10**, e61268 (2021).
5. Hochrein, L., Mitchell, L. A., Schulz, K., Messerschmidt, K. & Mueller-Roeber, B. L-SCRaMble as a tool for light-controlled Cre-mediated recombination in yeast. *Nat. Commun.* **9**, 1–10 (2018).
6. Roquet, N., Soleimany, A. P., Ferris, A. C., Aaronson, S. & Lu, T. K. Synthetic recombinase-based state machines in living cells. *Science (80-.)*. **353**, (2016).
7. Guo, L. *et al.* Engineering Escherichia coli lifespan for enhancing chemical production. *Nat. Catal.* **3**, 307–318 (2020).
8. Sheth, R. U. & Wang, H. H. DNA-based memory devices for recording cellular events. *Nat. Rev. Genet.* **19**, 718–732 (2018).
9. Sheets, M. B., Wong, W. W. & Dunlop, M. J. Light-inducible recombinases for bacterial optogenetics. *ACS Synth. Biol.* **9**, 227–235 (2020).
10. Lee, M. E., Deloache, W. C., Cervantes, B. & Dueber, J. E. A Highly Characterized Yeast Toolkit for Modular, Multipart Assembly. (2015). doi:10.1021/sb500366v
11. Benzinger, D. & Khammash, M. Pulsatile inputs achieve tunable attenuation of gene expression variability and graded multi-gene regulation. *Nat. Commun.* **9**, 1–10 (2018).

12. Zhao, E. M. *et al.* Optogenetic Amplification Circuits for Light-Induced Metabolic Control. *ACS Synth. Biol.* **10**, 1143–1154 (2021).
13. Hochrein, L., Machens, F., Messerschmidt, K. & Mueller-Roeber, B. PhiReX: A programmable and red light-regulated protein expression switch for yeast. *Nucleic Acids Res.* **45**, 9193–9205 (2017).
14. An-adirekkun, J. *et al.* A yeast optogenetic toolkit (yOTK) for gene expression control in *Saccharomyces cerevisiae*. *Biotechnol. Bioeng.* **117**, 886–893 (2020).
15. Xu, X. *et al.* A single-component optogenetic system allows stringent switch of gene expression in yeast cells. *ACS Synth. Biol.* **7**, 2045–2053 (2018).
16. Castle, S. D., Grierson, C. S. & Gorochowski, T. E. Towards an engineering theory of evolution. *Nat. Commun.* **12**, 1–12 (2021).
17. Din, M. O. *et al.* Synchronized cycles of bacterial lysis for in vivo delivery. *Nature* **536**, 81–85 (2016).
18. Balagaddé, F. K. *et al.* A synthetic *Escherichia coli* predator-prey ecosystem. *Mol. Syst. Biol.* **4**, 187 (2008).

Reviewers' Comments:

Reviewer #2:

Remarks to the Author:

I am satisfied with the authors' responses to my questions and would like to thank them for their extra work. I am happy to recommend this work for publication.

Response to reviewers

Reviewer #2 (Remarks to the Author):

I am satisfied with the authors' responses to my questions and would like to thank them for their extra work. I am happy to recommend this work for publication.

We thank the reviewer for his appreciation of our work.